# The Financial Burden of Tuberculosis for Patients in the Western-Pacific Region

**DOI:** 10.3390/tropicalmed4020094

**Published:** 2019-06-17

**Authors:** Kerri Viney, Tauhidul Islam, Nguyen Binh Hoa, Fukushi Morishita, Knut Lönnroth

**Affiliations:** 1Centre for TB Research, Department of Public Health Sciences, Karolinska Institutet, 17177 Stockholm, Sweden; knut.lonnroth@ki.se; 2Research School of Population Health, Australian National University, Canberra 2600, Australia; 3End TB and Leprosy Unit, World Health Organization Regional Office for the Western Pacific, 1000 Manila, Philippines; islamt@who.int (T.I.); morishitaf@who.int (F.M.); 4Vietnam National TB Programme, Ministry of Health, Hanoi 124302, Vietnam; nguyenbinhhoatb@yahoo.com; 5Centre for Operational Research, International Union Against Tuberculosis and Lung Disease, 75006 Paris, France

**Keywords:** tuberculosis, catastrophic costs, Western Pacific Region, social protection

## Abstract

The End Tuberculosis (TB) Strategy has the ambitious goal of ending the global TB epidemic by the year 2030, which is aligned to the Sustainable Development Goals. One of three high level indicators of the Strategy is the “catastrophic costs” indicator, which aims to determine the proportion of TB-affected households that incur TB-care related costs equivalent to 20% or more of their annual household income. The target is that zero percentage of TB-affected households will incur catastrophic costs related to TB care by the year 2020. In the Western Pacific Region of the World Health Organization, it is a priority to determine the financial burden of TB and then act to mitigate it. To date, eight countries in the Region have conducted nationally representative TB patient cost surveys to determine the costs of TB care. The results from four countries that have completed these surveys (i.e., Fiji, Mongolia, the Philippines, and Vietnam) indicate that between 35% and 70% of TB patients face catastrophic costs related to their TB care. With these results in mind, significant additional efforts are needed to ensure financial risk protection for TB patients, expand Universal Health Coverage, and improve access to social protection interventions. A multi-sectoral approach is necessary to achieve this ambitious goal by the year 2020.

## 1. Introduction

The End Tuberculosis (TB) Strategy has the ambitious goal of ending the global TB epidemic by the year 2030, which is aligned to the Sustainable Development Goals (SDGs) [1]. To measure the progress against this ambitious goal, the Strategy has three main high level indicators. These include a reduction in TB incidence rate by 80%, a reduction in number of TB deaths by 90%, and ensuring that no TB-affected household experiences “catastrophic costs” associated with TB diagnosis and care, which are all to be achieved by 2030, relative to baseline estimates in 2015 [1]. Of these three indicators, estimated TB incidence and mortality have been reported by the World Health Organization (WHO) on an annual basis since 1997 [2]. The third of these indicators is the newly defined “catastrophic costs” indicator, which was included in the End TB Strategy in recognition of the fact that the financial barriers to TB care are substantial and will likely impede achievement of global TB elimination goals [1]. In fact, the choice of this indicator as one of three high level indicators in the End TB Strategy acknowledges that the global epidemiological targets of reductions in mortality and incidence cannot be achieved without major progress towards Universal Health Coverage (UHC) and expanded access to social protection [1]. 

The financial costs to patients of a TB diagnosis and subsequent care can include medical costs (such as medical consultation fees), non-medical costs (such as transportation to get to the hospital), and indirect costs, such as time spent away from work, loss of income due to unemployment, or carer time. Previous studies have documented that TB patients often incur large costs related to their illness, including for a TB diagnosis and then for subsequent TB care [3,4]. A systematic review, which assessed the results of 49 studies on TB patient costs, concluded that these costs ranged from US $55 to $8198 (unweighted average of US $847) [3]. Income loss comprised the greatest proportion of all costs at 60% (range: 16–94%), with another 20% (range: 0–62%) due to direct medical costs [3]. The remaining 20% (range: 0–84%) of costs were due to direct non-medical expenses [3]. Half of the costs were incurred prior to the commencement of TB treatment [3]. Importantly, the total costs amounted to 58% (range: 5–306%) of annual individual income and 39% (range: 4–148%) of annual household income [3]. Costs in relation to income were higher for patients with lower incomes and also for people with multi-drug resistant TB (MDR-TB) [3].

Such costs create barriers to health care access and treatment adherence, which in turn can affect treatment outcomes [5] and prolong TB transmission. For example, in one study from Peru, those who experienced catastrophic costs were more likely to have an adverse treatment outcome, including treatment failure and recurrence [5]. These costs can also be detrimental to the socio-economic situation of TB affected households, especially for people who are already poor [3]. Measurement of the “catastrophic costs” indicator should enable governments to address demand-side cost barriers, which may be mitigated through a range of interventions including improving financial access to care, achievement of UHC, extending patient-centered care delivery models that reduce time needed for care-seeking, social protection interventions to mitigate loss of earnings due to care-seeking, and other forms of financial protection in times of illness. 

The measurement of TB patient costs using a standardized methodology for nationally representative surveys developed by WHO is currently being carried out by Ministries of Health in collaboration with WHO and other technical partners. The purpose of this review paper is to discuss the catastrophic costs indicator, to review previous evidence on the costs of TB care, to discuss current efforts to measure the financial burden of TB diagnosis and care, and to discuss potential policy responses to mitigate costs, highlighting the role of cost mitigation measures in achieving the ambitious goals articulated in the End TB Strategy. We conducted a narrative literature review where we searched PubMed and the grey literature for published information on studies that assessed the costs of TB care from the patient’s perspective in Asia-Pacific. Our review of the evidence on the costs of TB care is focused on the Western-Pacific Region of WHO (comprising 37 countries and areas). 

### 1.1. What are Catastrophic Costs in the Context of the End TB Strategy?

In the context of the End TB Strategy, catastrophic costs are defined by WHO as the total of “medical and non-medical out-of-pocket payments and indirect costs exceeding a given threshold (e.g., 20%) of the household’s income.” [1] The indicator is considered by WHO as “a key marker of financial risk protection (one of the two key elements of UHC) and social protection for TB affected households.” [1] 

Medical costs are defined as “out of pocket payments for TB diagnosis and treatment made by TB patients”, while non-medical costs are out of pocket “payments related to the use of TB health services, such as payments for transportation, accommodation or food.” [1] Collectively, these costs are referred to as direct costs. Indirect costs refer to “patient or guardian lost time, lost wages (net of welfare payments) and lost income due to TB health care seeking and hospitalization during TB care.” [1] The catastrophic costs indicator sums direct and indirect costs, so includes lost income and lost time, as well as the direct costs of care. The catastrophic costs indicator consists of a numerator, which is the number of people treated for TB (and their households) who incur catastrophic costs (direct and indirect combined), and a denominator, which is the total number of people treated for TB [1]. The threshold for determining if costs are catastrophic is 20% of annual household income. *The End* TB Strategy has a target that zero percent of TB patients and their households will experience catastrophic costs due to TB, to be achieved already by the year 2020 [1]. Aligned to the End TB Strategy, the Regional Framework for Action on Implementation of the End TB Strategy in the Western Pacific, 2016–2020 also has a specific target that zero TB-affected families will face catastrophic costs due to TB, which is to be achieved in the Region by the year 2020 [6]. In this regard, the Regional Framework calls for specific actions to minimize the “physical, financial and social barriers” associated with TB care [6]. 

To measure the proportion of TB patients with catastrophic costs, WHO recommends that periodic cross-sectional surveys of a nationally representative (random) sample of TB patients are required [7]. The surveys usually use a cluster sampling design, although random sampling of individual TB patients can also be used. TB patients who are reported to the national TB program are included, and the surveys include both adults and children and those with all types of TB. The 30 high TB burden countries have been prioritized for implementation of these surveys globally. WHO has published a handbook on the design, implementation, and interpretation of such surveys and has assembled a pool of experienced consultants who can assist countries to design, implement and analyze their surveys and assist with policy dissemination [7]. A generic survey protocol, questionnaire, and data analysis plan are also available, ensuring standardized measurement and analysis, allowing comparability across and within countries over time. Details on the methodology of national TB patient cost surveys is provided in WHO’s publication ‘Tuberculosis patient cost surveys: A handbook’ [7]. 

### 1.2. How Does “Catastrophic Costs” Differ from “Catastrophic Health Spending”?

The End TB Strategy catastrophic costs indicator differs substantially from the SDG indicator 3.8.2 on large (or catastrophic) health spending. The SDG indicator is defined as the “proportion of the population with large household expenditure on health as a share of total household expenditure or income [8]. Two thresholds are used to define “large household expenditure on health”: greater than 10% (SDG 3.8.2_10) and greater than 25% of total household expenditure or income (SDG 3.8.2_25) [8]. The SDG indicator is designed to capture “out-of-pocket” expenses on health and then to determine the impact of this expenditure on a household’s overall budget [8]. If these expenses are large, families may need to choose between spending money on health care versus other essentials such as food and education. In addition, this out-of-pocket health expenditure can cause households to incur catastrophic health spending, which in turn can push these households into poverty.” [8]

While there are a number of differences between the SDG indicator and the End TB Strategy indicator, the main differences are that the SDG indicator is primarily concerned with out-of-pocket expenses for health care and does not take into account opportunity costs such as lost income, unemployment or carer time, which are all captured in the End TB Strategy indicator. In addition, “the denominator of the SDG indicator includes many people who had no contact with the health system and thus had zero expenditures on health.” [2] Further, the SDG indicator is concerned with health expenditure for all illnesses and health concerns, whereas the End TB Strategy indicator focuses on costs related to TB care only, for the period from the onset of TB-related symptoms to completion of TB treatment. Therefore, the End TB Strategy indicator includes a wider array of costs but focuses on a single disease, i.e., TB. Due to the fact that the End TB Strategy indicator focuses on a specific patient group and that it includes indirect costs, it is expected that the proportion of TB patients facing catastrophic costs will be higher than the proportion of a population faced with catastrophic health spending. For this reason, while both indicators have utility in determining financial catastrophe related to health expenditure, they are not directly comparable. Table 1 highlights the differences between the catastrophic health spending, catastrophic costs related to TB care, and the UHC service coverage index for four countries who have reported their TB patient cost survey data to WHO in 2018. The UHC service coverage index is a measure of SDG indicator 3.8.1 (coverage of essential health services), which combines 16 tracer indicators of service coverage into a single summary measure [9]. It is defined as “the average coverage of essential services based on tracer interventions that include reproductive, maternal, newborn and child health, infectious diseases, non-communicable diseases and service capacity and access, among the general and the most disadvantaged population).” [10] The correlation or relationship between the UHC service coverage index indicator and catastrophic costs has not yet been studied. 

## 2. Costs of TB Care in the Western Pacific Region

### 2.1. Costs of TB Care in Studies not Using the WHO Methodology to Measure Catastrophic Costs

There is scant published literature on the costs of TB care for patients and their families in the countries of the Western Pacific Region. Aside from the national TB patient cost surveys, there is a limited number of published studies on this topic. A small number of studies have been published from China, Vietnam, and Cambodia [11,12,13,14,15,16,17,18]. Another paper from Australia reported on dissemination of vouchers in times of financial distress [19]. These studies used varied methodologies, so they are not directly comparable to each other nor are they comparable to the results currently being obtained from national TB patient cost surveys based on the WHO methodology. However, in one study from Cambodia, the tool used to measure TB patient costs was almost identical to the tool that WHO recommends currently [16]. Collectively, these studies provide very useful insights into the financial challenges faced by TB patients and their households, in China, Vietnam, and Cambodia. 

In one of these studies, the authors used a narrative review approach to assess how much patients paid for TB care in China [12]. Their objective was to determine if there was any association between the medical costs of TB care and treatment adherence [12]. This review identified nine studies from various provinces in China that provided estimates of total medical costs associated with TB treatment (excluding costs for transport and food) [12]. The medical costs of care ranged from ¥1241 (US $149) to ¥5228 (US $724), representing 39% to 119% of annual household income (an average of 63% of annual household income for six studies) [12]. In five studies that included quantitative data, 3–45% of non-adherence was attributable to the costs of TB care [12]. 

Another qualitative study conducted in Yunnan Province found that financial concerns were identified by TB patients and health care professionals alike as the “most pervasive barrier to care” [11]. These barriers were greater for patients with MDR-TB and for those from ethnic minorities and rural areas [11]. One interviewed TB patient described borrowing ¥6000 (US $942) for their first hospitalization, ¥5000 (US $785) for the second hospitalization, then the patient sold or borrowed assets worth ¥14,000 (US $2080) including the household’s motorbike [11]. The patient concluded by saying that their buffalo (worth ¥4000–5000; US $594–785) would be the next asset to be sold to finance TB care [11]. 

In another study from Jiangsu Province, 316 TB patients were interviewed as part of a cross sectional survey. The authors found that the average total out-of-pocket costs was ¥3024 (US $450) per capita [15]. This study measured out-of-pocket medical and non-medical costs and found these to be ¥2565.7 (US $318) and ¥458.3 (US $68), respectively [15]. In addition, the cost of productivity losses was ¥2615.2 (US $388) per person [15]. Factors associated with these costs included hospitalization, adverse drug reactions, administration of second line TB medicines, use of “liver-protecting” medicines, and costs associated with diagnostic delay [15]. Another quantitative study conducted in Jiangsu Province found that the average cost of inpatient TB care for 221 hospitalized patients was ¥13007.91 (US $1933; ± ¥5205.58) [14]. 

In another observational study from China that included 243 MDR-TB patients from four hospitals across the country (in Chongqing Municipality, Henan Province, Hohhot City, and Jiangsu Province), the authors determined the proportion of patients who experienced catastrophic costs under a new model of universal coverage for MDR-TB [17]. The method of calculating catastrophic costs was different to the current method that WHO is using, but in this study, the proportion of patients with catastrophic costs was 78%, despite 90% reimbursement of medical fees [17]. The authors conclude that enrolment in MDR-TB care would be higher if 100% of medical fees were reimbursed but that additional enablers would be needed to truly eliminate catastrophic costs [17]. 

A qualitative study from Vietnam found that indirect costs related to cumbersome diagnostic pathways and directly observed treatment in the public sector was an important reason for patients to instead seek care in the private sector despite higher direct medical costs there [20]. In another study from Vietnam, the authors used a cross-sectional survey design and interviewed 2014 MDR-TB patients from three tertiary care hospitals across Vietnam [13]. The authors found that the average cost of treatment for MDR-TB was US $1480.34 (± US $211.61) and US $2695.58 (± US $294.98) for patients receiving the 9- and 20-month MDR-TB treatment regimens, respectively [13]. Direct medical costs comprised the highest proportion of overall costs, with medicines accounting for a large proportion of these medical costs [13]. 

In one of the studies from Cambodia, the study investigators conducted interviews with TB patients from 17 health facilities to determine the financial burden of TB and to examine treatment outcomes associated with different forms of supervised treatment [21]. The costs of care varied according to the type of TB and model of care provided, from US $1900 for patients with non-multi-drug resistant TB receiving in-patient Directly Observed Therapy (DOT) to US $395 for DOT provided in the patient’s home; treatment outcomes were not statistically different for these two groups [21]. In Cambodia, in a cross-sectional comparative study, 208 structured interviews were held, 108 with TB patients who had been detected through active case finding and 100 through passive case finding [16]. The authors wanted to determine the costs of accessing TB care by comparing these two groups [16]. They found that active TB case finding reduced the median costs associated with TB care by 17% (US $240.7 [inter quartile range (IQR) 65.5–594.6] for patients detected through active case finding versus US $290.5 [IQR 113.6–813.4] for patients detected through passive case finding (*p* = 0.104)) [16]. Overall, the costs of TB care constituted 11.3% and 18.6% of annual household income for patients detected through active versus passive case finding, respectively [16]. 

In the one study from Australia, catastrophic costs were not determined; however, the author describes that one third of TB patients in the state of Victoria required emergency vouchers and that these vouchers were more likely to be distributed at the start of treatment, with funds used to relieve financial distress [19].

While the studies from China, Vietnam, and Cambodia highlight the financial challenges faced by TB patients in the region, there is scant published literature on the costs of TB care from the other 34 countries in the Western Pacific Region. 

### 2.2. Measuring the Costs of TB Care Using Nationally Representative TB Patient Cost Surveys

Globally, by December 2018, 13 countries had completed a national TB patient cost survey using the methodological approach recommended by WHO, including five countries from the Western Pacific Region [2]. By April 2019, a total of eight countries in the Western Pacific Region had either completed a national TB patient cost survey or were in the process of implementing one. These countries are China, Fiji, Laos, Mongolia, Papua New Guinea, the Philippines, Solomon Islands, and Vietnam. Japan and Malaysia are currently planning a national TB patient cost survey. A summary of national TB patient cost surveys in the Western Pacific Region and their stages of progress is provided in Figure 1.

Four countries that have conducted a national TB patient cost survey in the Western Pacific Region were ready to report their results to WHO in 2018 as a part of routine annual reporting. WHO has been requesting that countries report this indicator as part of annual reporting (if data are available) since 2017 [1]. The four countries that were able to provide such data for the Global Tuberculosis Report 2018 were Fiji, Mongolia, the Philippines, and Vietnam. A summary of the results from these countries is provided in Figure 2 below; the proportion of TB patients and their households experiencing catastrophic costs ranged from 35% in the Philippines to 70% in Mongolia [2]. The main drivers of these costs vary by country (and even within countries, according to the region or sub-group) [22]. In the Philippines and Vietnam for example, the main drivers of costs were direct non-medical costs and indirect costs, while in Mongolia, direct medical costs and direct non-medical costs were the main drivers of cost for drug-susceptible and drug-resistant TB patients, respectively [22].

Vietnam has recently published the results of their national TB patient cost survey in the peer reviewed scientific literature [23]. In the Vietnam national TB patient cost survey, a total of 735 TB patients participated from 20 sites across the country [23]. While 63% of TB patients experienced catastrophic costs overall, this figure was higher in the poorest quintile (83%) compared to the wealthiest (41%) and was almost universal for patients with MDR-TB, at 98% [23]. The difference in the proportion of households with catastrophic costs, comparing the poorest and wealthiest quintiles was primarily accounted for by lost income [23]. The mean costs for TB care were US $1054 and US $4302 per treatment episode for patients with drug-susceptible and MDR-TB, respectively, and US $1314 overall [23].

Other countries in the Region who have conducted national TB patient cost surveys are in the process of compiling reports and publishing their data; however, national dissemination of results has already begun, including to stakeholders outside of the health sector.

## 3. Program and Policy Implications

Results from nationally representative TB patient cost surveys provide important information for programming and policy development. This information is relevant to the social welfare and employment sectors, in addition to the health sector. In the Western Pacific Region—and globally—the results obtained from these surveys can inform policy and programming in a number of ways. Firstly, the costs of TB care “can be mitigated by improving approaches to TB service delivery and financing, such as removal of user fees and the introduction of more patient-centered models of care”, appropriate to the context [2]. The expansion and strengthening of UHC should facilitate this aim, as UHC aims for equitable access to health care services based on need and without financial hardship [24]. However, as UHC is primarily managed by the health sector, it may be necessary to move beyond UHC to mitigate remaining costs [25]. For example, indirect costs such as loss of income may not be mitigated through the expansion of UHC. Therefore, any remaining costs after optimization of health-care service delivery will need to be mitigated through improved social protection measures such as paid sick-leave, disability pensions, and other forms of income protection and legislation that prevents unfair dismissal from employment related to illness [2,25]. The two studies from Cambodia have demonstrated that the models of case finding and subsequent TB care can significantly impact on the costs faced by TB patients and provide important policy lessons on how to mitigate costs for TB patients [16,21]. Additional research is also underway in Vietnam to improve access to social health insurance (personal communication: N. V. Nhung). A summary of the possible policy implications of the results of the national TB patient cost surveys is provided in Table 2. WHO encourage the sharing of survey results with the relevant multi-sectoral stakeholders to facilitate policy uptake. (1) The results of national TB patient cost surveys can be used to stimulate discussion about UHC and financial and social risk protection as part of these multi-stakeholder consultations. 

In the Western Pacific Region specifically, after the results of the Vietnam TB patient cost survey were released, the Vietnamese Government convened a multi-sectoral meeting that was used to disseminate the findings of the survey and “formulate a joint action plan with the country’s Ministry of Labour and Social Affairs.” [2] A summary of the policy dialogue and actions arising from the Vietnam national TB patient cost survey is provided in Box 1. Major policy changes include the development and costing of a TB service package to advocate for patients to be covered by social health insurance, the launch of a charity fund for TB patients, and strengthening of the collaboration between the Ministry of Health and the Ministry of Labour-Invalids and Social Affairs.

It is important to note that stakeholder meetings convened before the implementation of a patient cost survey may aid dissemination after the survey has been completed. In Fiji, prior to the implementation of the survey, staff from the WHO Division of Pacific Technical Support and the Fiji Ministry of Health and Medical Services convened a stakeholder meeting to brief representatives from other government ministries, United Nations agencies, the Reserve Bank, Fiji National University, the Global Fund Country Co-ordinating Mechanism, non-governmental organizations, and others about the proposed TB patient cost survey prior to its implementation [7]. The objectives of this meeting were twofold: first, to provide information to interested parties on impending survey implementation; and second, to ensure early engagement with policymakers and others who can effect policy change so that survey results may be more easily considered and adopted by policy-makers [7]. The Ministry of Health and Medical Services plans to convene the same group of stakeholders when the survey results are available in order to disseminate the results and discuss policy implications.

Box 1**Results dissemination meeting and policy dialogue for the Vietnam national TB patient cost survey (Hanoi, March 2017).** The Vietnam National TB Programme has been sharing the study results of their national TB patient cost survey in a range of international conferences and forums, such as: (i) the 47th and 48th Union World Conferences on Lung Health (2016 and 2017); (ii) the first Social Protection Action Research & Knowledge Sharing network consultation, Karolinska Institutet, Stockholm, Sweden (2016); (iii) the 11th National TB Programme Managers Meeting in the Western Pacific Region (2017); and (iv) the WHO Global Task Force on TB Impact Measurement (2018). The Vietnam National TB Programme also collaborated with WHO in organizing a dissemination workshop in Hanoi, Vietnam, in March 2017, to share the results of the national TB patient cost survey; to discuss and identify key areas for policy actions; to develop a framework for monitoring and evaluation; and to undertake operational research on new policies, interventions, and approaches. In the workshop, the National TB Programme developed a roadmap (2017–2020), involving non-health actors, which outlines the policies and interventions to reduce and compensate for costs faced by TB patients and their households. The roadmap was designed in collaboration with the Ministry of Labour-Invalids and Social Affairs and other partners. The major components of the roadmap include (i) the development and costing of a TB service package service to advocate for patients to be covered by social health insurance; (ii) the launch of a charity fund for TB patients; (iii) strengthening of the collaboration between the Ministries of Health and Labour-Invalids and Social Affairs; and (iv) advocating for donor support to defray the costs of TB care for patients and their households.

## 4. Conclusions

Tuberculosis care, while lifesaving, can often be costly to patients. The End TB Strategy has an ambitious target that zero TB affected families will face catastrophic costs related to their TB care, to be achieved by the year 2020. Eight countries in the Western Pacific Region have conducted nationally representative TB patient cost surveys to determine the economic burden of TB; up to 70% of patients face catastrophic costs related to their care. Therefore, the achievement of this target is not likely to be met unless significant additional efforts to provide social and financial risk protection are put in place. Urgent, additional efforts are needed to reduce these costs by expanding access to UHC and by improving financial and social protection interventions. 

## Figures and Tables

**Figure 1 tropicalmed-04-00094-f001:**
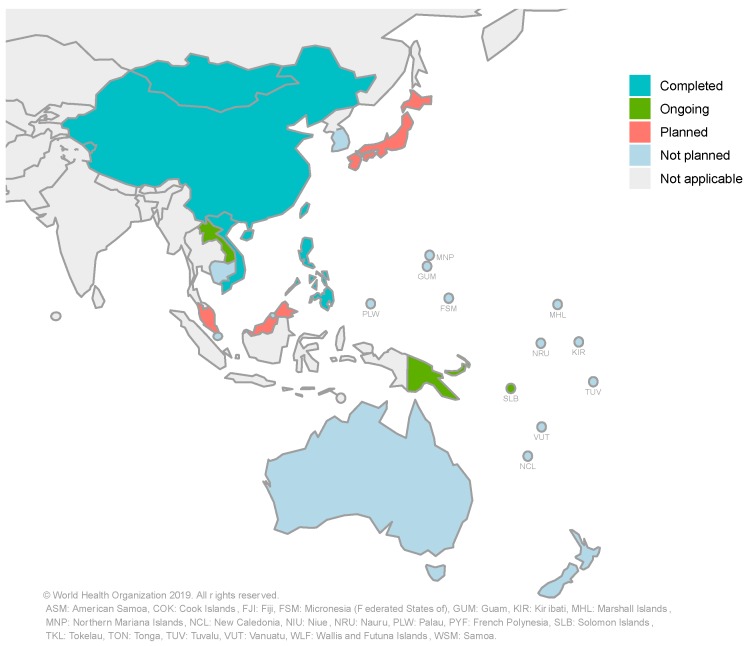
National surveys of costs faced by tuberculosis (TB) patients and their households in the Western Pacific Region: current progress and plans (updated in May 2019).

**Figure 2 tropicalmed-04-00094-f002:**
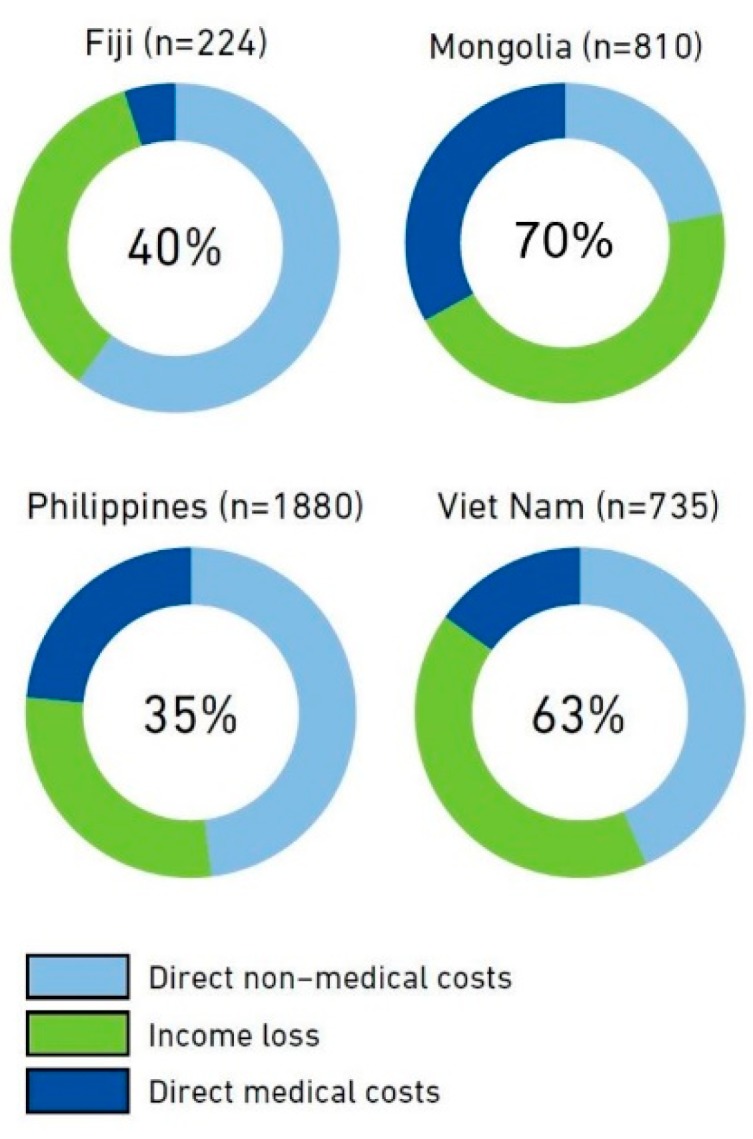
Results from the Fiji, Mongolia, Philippines, and Vietnam national surveys of costs faced by TB patients and their households, implemented 2016–2018. The number in the center of the circle is the best estimate of the percentage of TB patients and their households that experienced total costs that were above 20% of their annual income (i.e., catastrophic total costs). The outer ring shows the distribution of costs in three major cost categories. The number of TB patients included in each survey is shown after the country name. Source: Global Tuberculosis Report 2018, World Health Organization.

**Table 1 tropicalmed-04-00094-t001:** Catastrophic out-of-pocket health, catastrophic costs related to tuberculosis care, and essential health services coverage index for Fiji, Mongolia, the Philippines, and Vietnam.

Country	Catastrophic Out-of-Pocket Health Spending ^a^ (SDG 3.8.2)	Catastrophic Costs Related to TB Care	Universal Health Coverage Service Coverage Index (SDG 3.8.1) ^b^
Fiji ^c^	3.37%	40%	66
Mongolia ^d^	2.39%	70%	63
Philippines ^e^	6.31%	35%	58
Vietnam ^f^	9.81%	63%	73

Sources: Global Tuberculosis Report 2018 (2018); Global Health Observatory Data Repository (Western Pacific Region: 2017–2018): World Health Organization. ^a^ The threshold is for greater than 10% of total household expenditure or income. ^b^ This index is presented on a scale of 0 to 100. ^c^ The Fiji figure is from 2002. ^d^ The Mongolia figure is from 2012 ^e^ The Philippines figure is from 2015. ^f^ The Vietnam figure is from 2014. SDG = Sustainable Development Goal(s); TB = tuberculosis.

**Table 2 tropicalmed-04-00094-t002:** Examples of main cost categories and possible interventions that might be considered to eliminate costs or mitigate impact of costs.

Cost Category	Possible Changes in Service Delivery	TB Patient Social Support and Social Protection Schemes
Direct medical: before TB diagnosis	Streamline the TB patient pathway: Understand and adapt to treatment-seeking behaviors Update and promote the national standard of TB diagnosis and eliminate irrational testing Extend access to rapid molecular diagnostics Effectively use of chest radiography Improve links with private sector providers using consistent policies (e.g., quality of care, free of charge) Intensify targeted case finding, including systematic screening for priority risk groups	Reduce/subsidize/eliminate out-of-pocket payments (OOPs): Increase insurance coverage (general) Reimburse OOP made by TB patients Regulate and eliminate informal fees Engage relevant actors in or outside TB to identify opportunities that can enable better access
Direct medical: after TB diagnosis	Expand free-of-charge or highly subsidized TB service package including TB medicines, ancillary drugs, monitoring of adverse events, preventive treatment: Promote integrated management of comorbidities and risk factors (HIV, diabetes, other lung diseases, tobacco smoking, harmful use of alcohol): Improve the quality of TB care: Update and promote the national standard of TB care with an emphasis on people-centered care Eliminate irrational treatment, hospitalization and testing	Reduce/subsidize/eliminate OOP: Increase insurance coverage for TB-related services Increase insurance coverage for relevant comorbidities and risk factors Regulate and eliminate informal fees Improve provider payment mechanism to avoid over-provision of services Explore social protection available for specific vulnerable groups and people with medical conditions
Direct non-medical	Advocate local health-seeking and for care models bringing services close to patients, including community- and workplace-based care: Improve the quality of nutritional advice and regulate irrational nutritional recommendations by health care providers (e.g., supplements)	Provide assistance via TB program: Cash transfer Specific allowances (e.g., food, transportation, etc.) by cash, voucher, or in-kind Expand the use of general social assistance schemes: Engage NGOs, civil society organizations and patient groups to ensure patient support suitable for the locality
Indirect costs (income loss)	Range of interventions to enable earlier diagnosis and patient-centered care delivery that minimize time spent seeking and receiving care (decentralization, shorter waiting times, fewer health care visits, avoid unnecessary hospitalization, etc.): Improve access to social services: Improve health workers’ knowledge on social protection schemes Seamless link between health and social offices (one-stop site) Engage civil society and community organizations and volunteers in non-health sectors (social work, charity, legal services, and volunteers)	Facilitate enrolment of eligible patients/households in existing social protection schemes: Social assistance for poor and vulnerable families Sickness/disability grant Cash or in-kind transfer program Advocate review and/or improvement of social insurance as income replacement during illness Legislate and/or enforce provisions related to social, economic, and labor rights to protect individuals during TB illness and care

Source: World Health Organization. Tuberculosis patient cost surveys: A handbook. 2017. Geneva, Switzerland: World Health Organization.

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
