# Peer review of "The Financial Burden of Tuberculosis for Patients in the Western-Pacific Region"

_tropicalmed, 2019, doi:10.3390/tropicalmed4020094_

Round 1
Reviewer 1 Report
Overall remarks:
This paper provides a narrative review of the current situation regarding catastrophic costs for TB patients in the WHO Western Pacific region.
The manuscript provides an interesting and well-written analysis of an important and timely issue, especially given the targets set by the current End TB Strategy and Sustainable Development Goals, and the seriousness of this issue for many TB patients.
I have some small comments regarding the paper:
The paper would benefit from a brief methods section detailing the sources the authors accessed in compiling their review.
I am aware of a few other studies (see below) from the Western Pacific region that discuss catastrophic costs, that would seem to fall within the scope of this paper. Could the authors comment on how or why papers were included/excluded (this comment may be addressed by the authors’ response to my above comment regarding a methods section).
Duan et al. 2019 BMJ Open 9(4): e026638
Jiang et al. 2019 Infectious Diseases of Poverty 8:21
Li et al. 2016 Infectious Diseases of Poverty 5:5
Ruan et al. 2016 Int J Tuberc Lung Dis 20(5): 638-44
Watts 2018 Trop Med Infect Dis 3: E126
(Note, I appreciate that the 2019 papers may simply be too recent to have been included in the present review)
Section 1 – Introduction: lines 62 – 63: this sentence would benefit from an example or more detail.
Table 1: I was interested to note that Vietnam had the highest UHC service coverage index, but also the highest catastrophic out-of-pocket health spending and second-highest catastrophic costs, while the Philippines had the lowest UHC service coverage index but also the lowest catastrophic costs, relative to the other countries. Could the authors provide any comment on the factors that may underlie these differences (e.g. health systems factors, differing approaches to TB care in these countries, differing demographic or access factors, etc.), especially given the similar cost drivers mentioned later in the paper for these two countries (see lines 258 – 259)?
Section 2 – Costs of TB care in the Western Pacific Region: lines 170 and 233 – 235: these sentences mention ‘scant published literature’ from the other 34 countries in the Western Pacific region. Are the authors aware of any studies from these 34 countries? If so, I feel it would be of value to provide a brief mention of their contribution to the literature, as I strongly agree with the authors that other studies can provide useful insight into costs impacting TB patients in the region, despite methodological differences between studies (lines 177 – 178).
Section 3 – This section does not appear – was Section 4 mis-numbered?
Section 4 – Programme and policy implications: line 315 – there seems to be an extra word here – should it be ‘TB service package’ or ‘TB package service’?
Author Response
Comments and Suggestions for Authors – Reviewer 1
Overall remarks: This paper provides a narrative review of the current situation regarding catastrophic costs for TB patients in the WHO Western Pacific region. The manuscript provides an interesting and well-written analysis of an important and timely issue, especially given the targets set by the current End TB Strategy and Sustainable Development Goals, and the seriousness of this issue for many TB patients. I have some small comments regarding the paper:
The paper would benefit from a brief methods section detailing the sources the authors accessed in compiling their review.
Response: We have added in some additional information about the methods (lines 79-81).
I am aware of a few other studies (see below) from the Western Pacific region that discuss catastrophic costs, that would seem to fall within the scope of this paper. Could the authors comment on how or why papers were included/excluded (this comment may be addressed by the authors’ response to my above comment regarding a methods section).
Duan et al. 2019 BMJ Open 9(4): e026638
Jiang et al. 2019 Infectious Diseases of Poverty 8:21
Li et al. 2016 Infectious Diseases of Poverty 5:5
Ruan et al. 2016 Int J Tuberc Lung Dis 20(5): 638-44
Watts 2018 Trop Med Infect Dis 3: E126
(Note, I appreciate that the 2019 papers may simply be too recent to have been included in the present review)
Response: Thank you. The 2019 papers were not identified in our search due to their recency of publication. However, they are important contributions to the literature. We downloaded the five papers above and checkced them all. We have included reference to three of them in our review (we did not include the Jiang and Duan papers as they are mainly about out of pocket health expenditure and catastrophic health expenditure). The additional papers have been referenced in line 179 and there is some additional text from the Australian paper in lines 179-180 and 249-252, and additional information about one of the Chinese papers (Ruan et al) in lines 215-222.
Section 1 – Introduction: lines 62 – 63: this sentence would benefit from an example or more detail.
Response: We have added in an example in lines 63-65, with a reference.
Table 1: I was interested to note that Vietnam had the highest UHC service coverage index, but also the highest catastrophic out-of-pocket health spending and second-highest catastrophic costs, while the Philippines had the lowest UHC service coverage index but also the lowest catastrophic costs, relative to the other countries. Could the authors provide any comment on the factors that may underlie these differences (e.g. health systems factors, differing approaches to TB care in these countries, differing demographic or access factors, etc.), especially given the similar cost drivers mentioned later in the paper for these two countries (see lines 258 – 259)?
Response: We are not sure of the factors underlying these differences and it is a bit difficult to speculate without unpacking this further. We would think that if the UHC service coverage index is higher then catastrophic costs may be lower (although lost income is not captured in the UHC service coverage index), so it is not really clear why there would be a high UHC coverage index and also high catastrophic costs. The only apparent difference that we are aware of is that in the Philippines the sampling approach for the catastrophic costs survey used individual random sampling whereas in Vietnam a cluster random sampling approach was used. We have added in one sentence into the discussion about this (lines 155-157).
Section 2 – Costs of TB care in the Western Pacific Region: lines 170 and 233 – 235: these sentences mention ‘scant published literature’ from the other 34 countries in the Western Pacific region. Are the authors aware of any studies from these 34 countries? If so, I feel it would be of value to provide a brief mention of their contribution to the literature, as I strongly agree with the authors that other studies can provide useful insight into costs impacting TB patients in the region, despite methodological differences between studies (lines 177 – 178).
Response: Apart from the literature that we have found, we are not aware of other papers on the specific topic of the costs of TB care to patients. It appears to be a small body of literature, although we have not done a systematic review on this and this may be a next step, i.e. to review the literature more systematically with all of the countries of WPRO included. We are aware of all of the national TB patient cost surveys conducted in WPRO according to the methodology recommended by WHO and we have included the published surveys in our paper and some data from the unpublished surveys. Considering this, we have not made any additional changes to our paper in this regard.
Section 3 – This section does not appear – was Section 4 mis-numbered?
Response: This was an error and the sections have been re-numbered (i.e. sections 3 and 4).
Section 4 – Programme and policy implications: line 315 – there seems to be an extra word here – should it be ‘TB service package’ or ‘TB package service’?
Response: This is a typo which has been corrected.

Reviewer 2 Report
The study by Viney et al entitled “The financial burden of tuberculosis for patients in the Western-Pacific Region” is a well written review article and deserve publication. It is addressing an important health problem relevant to developing world. This article has discussed catastrophic costs indicator, evidence on the costs of TB care, current efforts to measure the financial burden of TB diagnosis and care and to discuss potential policy responses to mitigate costs and highlighted the role of cost mitigation measures in achieving the ambitious goals in the End TB Strategy. Article is publishable without any revisions/modifications.
Author Response
Response: Thank you for these comments.
Reviewer 3 Report
This study covers an important topic it is often not considered in the fight against tuberculosis. It provides an updated view on the costs of TB care on the Western-Pacific Region.
Author Response
Response: Thank you for these comments.